# Implementation of Exome Sequencing in Clinical Practice for Neurological Disorders

**DOI:** 10.3390/genes14040813

**Published:** 2023-03-28

**Authors:** María Isabel Alvarez-Mora, Laia Rodríguez-Revenga, Meritxell Jodar, Miriam Potrony, Aurora Sanchez, Celia Badenas, Josep Oriola, José Luis Villanueva-Cañas, Esteban Muñoz, Francesc Valldeoriola, Ana Cámara, Yaroslau Compta, Mar Carreño, María Jose Martí, Raquel Sánchez-Valle, Irene Madrigal

**Affiliations:** 1Biochemistry and Molecular Genetics Department, Hospital Clinic of Barcelona, IDIBAPS (Institut de Investigacions Biomèdiques August Pi I Sunyer), 08036 Barcelona, Spain; 2CIBER of Rare Diseases (CIBERER), 08036 Barcelona, Spain; 3Molecular Biology of Reproduction and Development Research Group, Department of Biomedical Sciences, Faculty of Medicine, Universitat de Barcelona, 08036 Barcelona, Spain; 4Molecular Biology CORE (CDB), Hospital Clínic de Barcelona, 08036 Barcelona, Spain; 5Parkinson’s Disease & Movement Disorders Unit, Neurology Service, Hospital Clínic Universitari de Barcelona, IDIBAPS, CIBERNED (CB06/05/0018-ISCIII), ERN-RND, Institut Clínic de Neurociències UBNeuro (Maria de Maeztu Excellence Centre), Universitat de Barcelona, 08036 Barcelona, Spain; 6Epilepsy Unit, Department of Neurology, Hospital Clinic, 08036 Barcelona, Spain; 7Neurology Department, Clinical Institute of Neurosciences, Hospital Clinic of Barcelona, 08036 Barcelona, Spain; 8Biomedical Research Institute August Pi i Sunyer (IDIBAPS), Hospital Clinic of Barcelona, 08036 Barcelona, Spain

**Keywords:** neurological disorders, whole exome sequencing, neurodevelopmental disorders, autism spectrum disorder, Parkinson, epilepsy, dystonia, ataxia, spastic paraplegia

## Abstract

Neurological disorders (ND) are diseases that affect the brain and the central and autonomic nervous systems, such as neurodevelopmental disorders, cerebellar ataxias, Parkinson’s disease, or epilepsies. Nowadays, recommendations of the American College of Medical Genetics and Genomics strongly recommend applying next generation sequencing (NGS) as a first-line test in patients with these disorders. Whole exome sequencing (WES) is widely regarded as the current technology of choice for diagnosing monogenic ND. The introduction of NGS allows for rapid and inexpensive large-scale genomic analysis and has led to enormous progress in deciphering monogenic forms of various genetic diseases. The simultaneous analysis of several potentially mutated genes improves the diagnostic process, making it faster and more efficient. The main aim of this report is to discuss the impact and advantages of the implementation of WES into the clinical diagnosis and management of ND. Therefore, we have performed a retrospective evaluation of WES application in 209 cases referred to the Department of Biochemistry and Molecular Genetics of the Hospital Clinic of Barcelona for WES sequencing derived from neurologists or clinical geneticists. In addition, we have further discussed some important facts regarding classification criteria for pathogenicity of rare variants, variants of unknown significance, deleterious variants, different clinical phenotypes, or frequency of actionable secondary findings. Different studies have shown that WES implementation establish diagnostic rate around 32% in ND and the continuous molecular diagnosis is essential to solve the remaining cases.

## 1. Background

Many neurological disorders have a monogenic Mendelian basis and affect genes involved in the normal function of the brain. The spectrum of such diseases ranges from the rare and early-onset neurodevelopmental disorders (NDD) to the more common late-onset neurodegenerative diseases. In the last two decades, the identification of genetic causes of neurological disorders has had a significant impact on the comprehension of pathological mechanisms. In addition, driven by the new sequencing possibilities and the genetic and phenotypic variability of many diseases, clinical genetic testing has changed drastically in the last decade. Prior to the NGS era, the study of intellectual disabilities (ID) was limited to Fragile X syndrome and copy number variants (CNVs) detected by chromosomal microarray or MLPA. Further examples are the genetic studies of spastic paraplegia and spinocerebellar ataxias, in which genetic interrogations was limited to the *SPAST* gene and discarding microsatellite expansions of typical mutated genes (SCA1, SCA2, SCA3, SCA6, SCA7, FXTAS, DRPLA, and FRDA), respectively.

Massively parallel sequencing technology has become the standard technique for clinical practice. Specifically, whole exome sequencing (WES) has led to enormous progress in deciphering monogenic forms of various genetic diseases [1,2,3]. WES offers the possibility of simultaneously testing for the presence of pathogenic variants in all known disease-related genes, improving not only the diagnostic process, but also making it faster and more efficient. Reaching a diagnosis in several rare genetic diseases is still a challenge, given the considerable clinical and genetic heterogeneity associated with them [4].

WES is widely regarded as the current technology of choice for diagnosing monogenic neurological disorders. In fact, in 2021, there has been an update of the current guidelines for genetic diagnosis of patients with developmental delay, ID, or congenital anomalies moving from Miller and collaborators (2010) proposing chromosomal microarray as a first-tier clinical diagnostic [5] to the new recommendations of the American College of Medical Genetics and Genomics (ACMG), which strongly recommend applying NGS as a first- (or second-) line test [6]. However, despite the revolutionary advance of genetic testing, a high percentage of rare genetic diseases lack for genetic diagnosis.

Since the implementation of the NGS in our laboratory, we have faced a continuous and exponential growth in the number of clinical exomes requested. In the last 3 years, we approximately sequenced 500–600 exomes per year in patients affected with several genetic diseases and a total of 209 exomes from patients with neurological disorders were analyzed at the end of 2022. The main aim of this report is to discuss the impact and advantages of the implementation of WES a routine tool for clinical diagnosis and management of neurological disorders. Therefore, we have performed a retrospective evaluation of WES application in 209 cases referred by neurologists or clinical geneticists to the Department of Biochemistry and Molecular Genetics of the Hospital Clinic of Barcelona for WES sequencing. In addition, we have further discussed some important facts regarding classification criteria for pathogenicity of rare variants, variants of unknown significance (VUS), and secondary findings.

## 2. Material and Methods

### 2.1. Patients

We have conducted a retrospective study of the application of NGS in 209 patients with clinical diagnosis of neurodevelopmental, neurological, or neurodegenerative disorders who were referred from January 2019 to December 2022 to the Department of Biochemistry and Molecular Genetics of the Hospital Clinic of Barcelona for genetic testing.

Patients encompass different clinical groups:Movement disorders cohort: 40 patients with ataxia ruled out microsatellite expansions (SCA1, SCA2, SCA3, SCA6, SCA7, DRPLA, and FXTAS), 38 patients with spastic paraplegia, 46 patients with dystonia, and 23 patients with Parkinson’s disease.NDD cohort: 20 patients with ID, 8 patients with autism spectrum disorder (ASD), and 23 patients with seizures. *FMR1* expansion and CNVs were previously discarded in all patients.Other disorders: 15 patients with other neurological conditions including microcephaly, leukodystrophy, neurological channelopathies, familial hemiplegic migraine, within others.

The institutional review board approved the collection and use of these samples for research purposes (Ethics Committee of Hospital Clinic of Barcelona 2011/6625). Written informed consent was obtained from all subjects or the guardians/parents of such patients, prior to their participation. DNA extraction from peripheral blood was performed with the MagNA Pure Compact Nucleic Acid Isolation kit using a Magna Pure 96 instrument (Roche Diagnostics).

### 2.2. Sequencing and Bioinformatic Analysis

Until 2021, WES libraries were prepared using the Illumina Nextera Flex for Enrichment assay (Illumina, San Diego, CA, USA) and subsequent paired-end sequencing (2 × 150 bp) of the whole exome was performed with the Nextseq550 platform (Illumina). From 2022, WES libraries were prepared using the DNA Prep with Enrichment (Illumina) and the SureSelect Human All Exon v8 kits (Agilent Technologies, Santa Clara, CA, USA). Paired-end sequencing (2 × 150 bp) of the whole exome was performed with the Illumina NextSeq 2000 sequencer (Illumina, San Diego, CA, USA). Briefly, bioinformatics analysis was done with an in-house pipeline (coreBM-Germline_1.0.0) based on nf-core/Sarek and following GATK best practices [7]. Read alignment was done against hg38 reference genome using bwa-mem and variants were called using GATK HaplotypeCaller v4.1.7.0. Filtering of genetic variants was performed according to the quality metrics of alignment and genotyping. Annotation of the variant calling files was done using the Jnomics platform (jnomics.es). Variants identified from alignments with poor mapping quality, variants with a frequency greater than 3% in any of the databases used (GnomAD and 1000 Genomes) for genes associated with recessive inheritance or a frequency greater than 0.5% for genes associated only with dominant inheritance and deep intronic variants with a reading depth of less than 10X and minimal coverage lower than 20% were not assessed.

WES data analysis was based on custom panels testing (available upon request) to focus attention on those genes most often associated with a specific disease (Hereditary Ataxia: 350; Spastic paraplegia: 114; Dystonia: 53; Parkinson: 79; ID: 1892; ASD: 1002; and Epilepsy: 677). These panels were designed by molecular geneticists from the Department of Biochemistry and Molecular Genetics of the Hospital Clinic and are revised and updated over the time.

Putative candidate variants were prioritized according to the predicted impact on coding sequence, their gene function, zygosity and genetic mode of inheritance, clinical features, and their presence in ClinVar [8] or the Human Gene Mutation Database (HGMD) [9].

## 3. Results and Discussion

### 3.1. Variant Classification

The challenge of the implementation of WES in routine diagnosis is to identify pathogenic variants from millions of unknown/uncertain significance (VUS) and benign variants. Recently the ACMG together with the Association for Molecular Pathology (AMP) and the College of American Pathologists (CAP), developed new variant interpretation guidelines [10] that are the standards used for variant classification in genetic diagnosis. These guidelines established classification criteria that presented different degrees of evidence of pathogenicity and benignity so that, through an algorithm that used the combination of these criteria, the variants could be classified as pathogenic (P), probably pathogenic (PP), with unknown clinical significance (VUS), probably benign (PB), or benign (B). Nevertheless, the application of this algorithm has not prevented the increase in VUS. There are currently multiple tools for variant classification and prioritization (Table 1). Some of these platforms have information about previous reports of the variant of interest and clinical associations. The genetic report should at least contain all relevant information used to interpret the result and variant classification, e.g., literature resources, reference sequences, and databases of normal variation, such as the Genome Aggregation Database (gnomAD; https://gnomad.broadinstitute.org (accessed on 21 March 2019), Database of Genomic Variants (DGV) (http://dgv.tcag.ca/dgv/app/home (accessed on 21 March 2019)), or disease-specific databases, including the Leiden Open Variation Database (LOVD) (https://www.lovd.nl (accessed on 21 March 2019)). Control and patient databases are of great help in order to classify detected variants. Population and disease databases should be frequently updated, and state what methods were used for data curation. Nevertheless, we must be aware that control databases cannot be assumed to include only healthy individuals. The number of ‘healthy controls’ in gnomAD are derived from various different cohorts, but there is no a real guarantee that all affected individuals have been excluded. An example of an erroneous classification might be the variant rs141976717 in the *NIPBL* gene, which a priori seems to be a VUS—likely benign variant. We demonstrated that this variant was originated de novo in a patient with growth retardation, intellectual disability, global cognitive and growth retardation, and microcephaly. Physical examination revealed obvious clinical signs of CdLs, such as long philtrum, long eyelashes, high-arched palate, anteverted nares, and hypospadias. Despite this variant being present in one individual in gnomAD v2 (https://gnomad.broadinstitute.org/variant/5-36961612-T-C?dataset=gnomad_r2_1 (accessed on 15 June 2021)) and in two individuals in gnomAD v3 (https://gnomad.broadinstitute.org/variant/5-36961510-T-C?dataset=gnomad_r3 (accessed on 15 June 2021)), and all the in silico predictors tools suggesting it was not deleterious, the absence of additional genetic alterations associated to NDDs and de novo origin reinforced the idea of a probably pathogenic variant [11]. Hence, when using a general population as a control cohort, the presence of individuals with subclinical disease is always a possibility and thus variants with a low allelic frequency should not be discarded.

Publicly available databases used for data analysis: gnomAD (www.gnomad.broadinstitute.org (accessed on 21 March 2019)), ClinVar (www.ncbi.nlm.nih.gov/clinvar (accessed on 21 March 2019)), LOVD (www.lovd.nl (accessed on 21 March 2019)), HGMD (www.hgmd.cf.ac.uk (accessed on 21 March 2019)), Uniprot (www.uniprot.org (accessed on 21 March 2019)), OMIM (www.omim.org (accessed on 21 March 2019)), Orphanet (www.orpha.net (accessed on 21 March 2019)), and GeneReviews (www.ncbi.nlm.nih.gov/books/NBK1116 (accessed on 21 March 2019)). Classification of the variants was performed according to the criteria established by the American College of Medical Genetics (ACMG) and Genomics based on the information extracted from the resources consulted [10].

### 3.2. Diagnostic Yield

WES application allowed the identification of disease-causing variants in 66 patients, representing an overall diagnostic yield of 32%. Appendix A shows the list of pathogenic or probably pathogenic variants detected in each patient. The data showed that according to the disease type, diagnostic rates range from 65% in spastic paraplegia to 15% in dystonia (Table 2, Figure 1). These differences could be related to the number of known genes associated with the disease, the probability of a non-Mendelian inheritance of the disorder (multiple alleles, incomplete dominance, digenic diseases), or the existence of new genes not identified yet. Similar diagnostic rates were obtained in the movement disorder cohorts and NDD cohorts.

Our results are in consonance to diagnostic yields reported by other groups in the context of clinical diagnostic testing (Table 2). The higher diagnostic yields obtained in spastic paraplegia and ID groups might reflect a rigorous patient selection due to the limitation of the number of cases studied per year.

### 3.3. Variants of Unknown Significance

The presence of VUS adds complexity to the interpretation of the results. In this cohort, we have found a similar percentage of VUS in all clinical groups, except in those patients with epilepsy (Figure 1). These results might reflect the fact that, besides classifying a variant as VUS, there are other factors that can complicate genetic diagnosis, such as the identification of variants in genes with incomplete penetrance or susceptibility genes.

Many variants are difficult to classify as benign or pathogenic, e.g., novel missense variants, novel putative splicing variants with unproven effect at the RNA level, among others. The ACMG has delineated definitions and guidelines for the interpretation of VUS [10]. In principle, these variants should not be used in clinical decision-making, unless they are later reclassified. In silico analysis and experimental laboratory studies could help in determining the potential therapeutic value of a VUS. Some tools include the prediction whether a missense change is damaging to the resultant protein structure or function, and those that predict if there is an effect on splicing. Nevertheless, functional studies for missense variants are usually difficult to implement in the routine diagnosis. Missense variants have variable effects upon protein function, ranging from loss of protein due to severe instability, to no discernible consequence, and interpreting their pathogenicity is challenging.

Different studies can be carried out by the clinical laboratory to deepen the interpretation of certain variants, with segregation analysis being the gold-standard. The most notable success of this type has been in identifying de novo variants in a wide variety of neurological disorders, including ID, epilepsy syndromes, ASD, among others. In addition, segregation analysis might lead to the reclassification of a VUS to LP variant in large families with multiple affected individuals. However, in smaller families, this method might still prove inconclusive. Functional studies, both in vivo or in vitro, are a powerful tool to determine the deleterious effect of the variant. Nevertheless, these types of studies are not available in most of routine diagnostic laboratories. In our set of patients, we only performed functional studies from those VUS with a high risk of affecting splicing. There are different predictive tools, such varSEAK (https://varseak.bio/, accessed on 21 March 2019) or Alamut (Alamut Visual Plus © Sophia GENETICS 2021), that predict the probable splicing effect of the variant (Table 1). Notwithstanding the above, it is recommended to reanalyze VUS variants at regular intervals in order to reclassify them on the basis of new evidence [20]. When a VUS is detected, it should always be recommended to perform a reanalysis of this variant at regular intervals since variants become reclassified over time on the basis of new evidence [20].

Another aspect to take into account is genes that have a defined spectrum of benign and pathogenic variation to identify genes subject to strong selection against various classes of variants. The predicted constraint metrics, Z-score and pLI score, are frequently used in order to prioritize candidate genes when analyzing WES or WGS data. Z-scores are available for the missense and synonymous categories. A greater Z-score indicates more intolerance to the class of variation. For genes in which missense variation is a common cause of disease and there is very little benign variation observed in the gene, a novel missense variant can be considered supporting evidence for pathogenicity. On the other hand, pLI scores are available for the loss-of-function variation. pLI closer to 1 indicates that the gene cannot tolerate protein truncating variation (nonsense, splice acceptor, and splice donor variation). For genes in which truncating variants are the only known mechanism of variant pathogenicity, missense variants can be considered supporting evidence for a benign impact.

### 3.4. Genes with Incomplete Penetrance and Variable Expressivity

The concepts of variable expressivity and incomplete penetrance are important factors in several genetic disorders that hamper genetic analysis in many of them. Monogenic genotypes can be highly predictive for specific individual disorders, but sometimes this relationship can be complicated. Individuals with the same genotype can display distinctly different clinical phenotypes [23,24,25], including being clinically asymptomatic. These cases present a challenge for clinicians, leading to uncertainty over whether a clinical phenotype will develop, and if so, when. Little is known about the penetrance of several variants in the genes implicated in ID and developmental delay. Research into the penetrance and expressivity of such genetic variants is important both for determining causative mechanisms of the disease and for providing accurate risk information through genetic counselling. The identification of other genetic, epigenetic, or environmental factors would clarify the pathogenesis of many diseases and may lead to better management and treatment of the disease. Moreover, the presence of putatively pathogenic variants in asymptomatic adults also highlights the possibility that there are disease resistance mechanisms, and these mechanisms could be identified through general population sequencing.

### 3.5. Secondary Findings

The introduction of whole exome or genome sequencing leads to massive genetic data. In routine clinical practice, only variants in genes that have already been considered pathogenic or probably pathogenic are routinely reported. Nevertheless, in some cases, patients who undergo genetic testing are found to have variants in genes that are unrelated to the disease described in the patient. Some of these findings could cause medical conditions that may be asymptomatic for long periods, could include carrier status for certain disorders, report pharmacogenetic variants, or could be useful for the prevention and treatment of other potential diseases [26,27,28]. The ACMG/AMP have published recommendations for reporting such findings for clinical evaluation [26,29]. In the 2014 version, secondary findings would only be informed if patients signed an informed consent. However, the 2017 version includes an option to opt-out of receiving secondary findings [26]. The European Society of Human Genetics (ESHG) suggested some specific recommendations based on aspects such as the risk–benefit balance, the costs of screening, the availability of preventive and therapeutic measures, respect for the principle of patient autonomy, the psychological and medical impact, and inequalities affecting access to health services [30]. The identification of such secondary findings can vary per cohort since diagnostic laboratories are not obliged to follow these guidelines. In our laboratory, secondary findings are only reported in genes associated with processable diseases, such as cystic fibrosis, or in allele carriers in autosomic recessive disorders with a frequency greater than 1/125.

### 3.6. Variants Nomenclature

A uniform nomenclature, informed by a set of standardized criteria, is recommended to ensure the unambiguous designation of a variant, and enables effective sharing. Nomenclature of detected genetic variants should be meaningful and consistent using the recommended Human Genome Variation Society (HGVS) (http://www.hgvs.org/mutnomen (accessed on 21 March 2019)) [31] and genome build and transcript references used for naming variants should be determined. The transcript should either represent the longest known transcript and/or the most clinically relevant transcript. Reference sequences must come from data sources that provide stable and permanent identifiers (e.g., RefSeq or Ensembl). A sequence identifier must only ever identify one reference sequence and all references sequence identifiers should use version numbers to distinguish between sequences. Variant descriptions lacking a version number are not valid. Depending on the variants to be reported, different reference sequence files are used. Indication of the reference type sequence file is mandatory. For DNA, the approved reference sequence types are g. (linear genomic reference sequence), c. (coding DNA reference sequence), n. (non-coding DNA reference sequence), m. (mitochondrial reference), and o. (circular genomic reference sequence).

### 3.7. WES Limitations and New Diagnostic Strategies

Despite numerous studies that show the efficacy of NGS in establishing molecular diagnoses, pathogenic variants are generally identified in less than 50% of patients with genetic neurological disorders. There may be several reasons why WES does not identify genetic causes. Some of them are technical, e.g., low capture efficiency or the presence of genomic regions that are difficult to sequence such as GC-rich regions, among others. Furthermore, some variant types, such as repeat expansions or structural variations, are still challenging to identify with short-read technologies. CNVs detection is still challenging in WES, although some algorithms are being improved to detect these variants. For example, we have recently detected a deletion implicating the *PMP22* gene in a patient with clinical suspicion of hereditary neuropathy with pressure palsies (not included in this cohort). Besides, variants occurring in non-coding sequences will not be detected by WES. Nevertheless, diagnostic rates could increase with the implementation of new diagnostic strategies, such as, for example, whole genome sequencing (WGS) or long read sequencing. Although the implementation of WGS in routine diagnosis remains challenging, the clinical utility of WGS in cases with only one heterozygous variant in recessive diseases may help to identify the second pathogenic variant [32]. Another new strategy is the long-read sequencing, which offers a number of advantages over short-read sequencing. This technology enables the detection of complex structural variants, identification of complete isoforms (RNAseq), or the detection of epigenetic marks, among other applications. Transcriptomic analyses are also of great help for classifying variants. When integrated with exome or genome sequencing, gene expression profiling has been shown to significantly improve molecular diagnosis rates [33,34,35]. This additional testing could greatly improve the diagnostic value, as studies have shown that synonymous variants and deep intronic variants can result in splicing and other RNA processing defects [36]. On the other hand, variant interpretation in clinical practice generally excludes the reporting of genes for which there is currently no known clinical relevance. Therefore, it is highly probable the existence of unidentified genes responsible for several genetic diseases. Nowadays, there are commonly used databases in order to link clinical and research groups that have identified rare variants in the same genes. One of these databases is GeneMatcher, an online tool designed to advance novel gene discovery in rare diseases [37]. Nevertheless, it is still complicated to perform WGS, transcriptomic, or methylation analysis in diagnostic labs. In our case, the majority of undiagnosed patients in our cohort are included in research projects.

### 3.8. Genetic Counselling

Bringing exome sequencing into clinical routine care is transformational for patients and families. Discovery of the etiology is important, as it often carries management implications, which can improve outcomes. It is not unusual for genetic counselling sessions to include a pre- and post-testing session. Among the topics that may be discussed during a pre-testing session are the clinical presentations of the condition, the pattern of genetic inheritance, collection of family medical history, available testing procedures, and test limitations. If the patient decides to have genetic testing performed, the genetic counsellor should obtain informed consent for any necessary genetic testing. The post-test session includes the provision of medical information and often focuses on helping families cope with the medical, emotional, and psychological consequences of the test results. If the genetic test is positive, general questions relating to suggested treatment or therapy are addressed. In some diseases, specific treatment can be addressed to the patient and the state anxiety levels of parents could decrease significantly after learning the diagnosis. Moreover, genetic diagnosis has great importance in estimating disease risk for family members. Genetic results often provide critical information for accurate reproductive counselling to young parents considering further children Information about community resources and support groups can be provided to the patient/family. Referrals may be made to specialists regarding specific issues that fall outside the scope of genetic counselling practice.

Genetic counselling is more complex for variants related to late onset diseases, since features associated with specific genetic diseases may not emerge until adult life. It is just as complicated when the variant is found in a gene responsible for a disease with incomplete penetrance, since it is not possible to know if and when the clinical symptoms may appear. The genetic counselling can be more complex in families in which the patient is the only individual affected and carries a variant of incomplete penetrance or is affected by a late-onset disease. In these cases, the detection of the variant in healthy individuals cannot rule out that the variant is not pathogenic.

## 4. Conclusions

Today, geneticists can routinely sequence at the whole genome level instead of performing the tedious process of sequencing and interrogating one gene at a time. Best practices for NGS interpretation in clinical practice require liaison with clinical geneticists and molecular diagnostic laboratories. We believe that continuous molecular diagnosis is essential to resolve the remaining cases. In this article, we have shared our experience in incorporating exome sequencing in the routine diagnosis of neurological disorders. In our cohort, WES application found molecular causes in 32% of the patients. Improvements in genetic diagnostics require generation of larger databases, both for pathogenic and benign variants, more data sharing, and more research for a deeper understanding of exome sequencing. On the other hand, one of the most important aspects for the patient is genetic counselling. In addition to receiving a genetic diagnosis of the disease, patients should receive all information related to the clinical presentation of the disease, the risk of the genetic pattern of inheritance of the disease, the possibility of recurrence, available test procedures, limitations reproductive options, and follow-up procedures, if necessary. Although the ability to perform functional analysis is still limited in several diagnostic labs, the implementation of new diagnostic tools will improve genetic diagnostic rate and will allow the resolution of undiagnosed cases.

## Figures and Tables

**Figure 1 genes-14-00813-f001:**
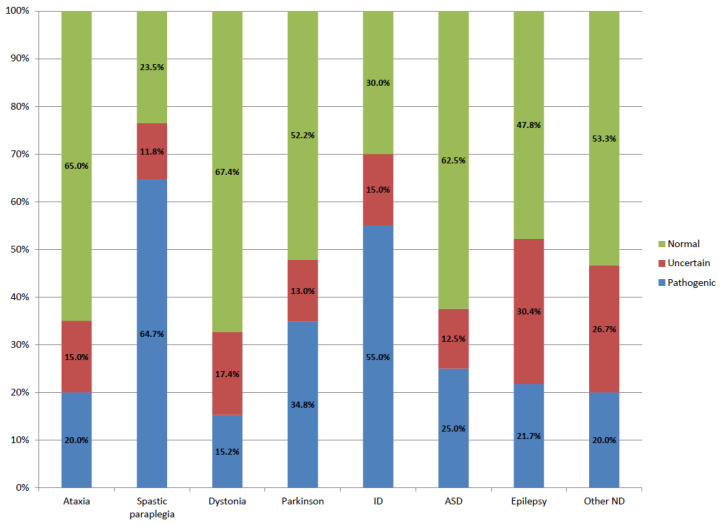
Diagnostic rate of WES in patients with various neurological diseases. Blue bars indicate the percentage of cases in which a pathogenic variant has been detected; red bars indicate cases in which a VUS has been detected; and green bars indicate cases in which no responsible variant has been detected.

**Table 1 genes-14-00813-t001:** Main tools for variant interpretation.

Tools for Variant Interpretation	Description	Examples
Databases of genomic variants	They report gene variants with information on their clinical involvement or bibliographic sources in which they are mentioned.	ClinVar, dbSNP ^1^, HGMD ^2^, LOVD ^3^, DGV ^4^, or LitVar
Predictive programs or in silico studies	These programs include the importance of the alteration both at the nucleotide level and at the amino acid level. They are divided in two groups: (1) prediction whether the change is detrimental to the function or structure of the resulting protein, and (2) prediction if splicing is altered.	PolyPhen2, SIFT, Alamut, or Mutation Taster GeneSplicer, Human Splice Finder, Alamut, REVEL ^5^ CADD ^6^, or varSEAK
Evaluation of the frequency of the variant in the control population	Databases of exome and genome sequencing data from a wide variety of large-scale sequencing projects. These databases describe and analyze human genetic variation.	gnomAD ^7^, 1000 Genomes, or ESP ^8^
Decision support software	These tools integrate information from several databases and combine it to carry out a classification according to the 2015 ACMG/AMP clinical guidelines	Franklin or Varsome

^1^ Single Nucleotide Polymorphism Database, ^2^ Human Gene Mutation Database, ^3^ Leiden Open Variation Database, ^4^ Data Base of Genomic Variants, ^5^ Rare Exome Variant Ensemble Learner, ^6^ Combined Annotation Dependent Depletion Exome, ^7^ Genome Aggregation Database; ^8^ Exome Sequencing Project.

**Table 2 genes-14-00813-t002:** Diagnostic yield in our cohort and other studies in patients with different neurological diseases.

Disease	Number of Patients Analyzed	Our Cohort % P/PP Variants	Other Reports % P/PP Variants
Movement disorders cohort			
Ataxia	40	20% (8/40)	13–52% [12,13,14,15,16]
Spastic paraplegia	34	64.7% (22/34)	40% [15]
Dystonia	46	15.2% (7/46)	8–37% [13,15,17,18]
Parkinson	23	34.8% (8/23)	11–14% [15,19]
Total		31.5% (45/143)	
Neurodevelopmental disorders cohort			
ID	20	55% (11/20)	22–48% [11,13,18,20]
ASD	8	25% (2/8)	9–21% [12,13,16,18,21]
Epilepsy	23	21.7% (5/23)	15–40% [12,13,16,18,22]
Total		35.3% (18/51)	
Other disorders			
Other	15	20% (3/15)	
ALL COHORTS	209	31.57% (66/209)	

P: pathogenic; PP: probably pathogenic; ID: intellectual disability, ASD: autism spectrum disorder.

## Data Availability

The data presented in this study are available in the Biochemistry and Molecular Genetics Department of the Hospital Clínic of Barcelona, Spain, under request.

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
