# Peer review of "Implementation of Exome Sequencing in Clinical Practice for Neurological Disorders"

_genes, 2023, doi:10.3390/genes14040813_

Round 1

Reviewer 1 Report

Alvarez-Mora et al., has performed WES in 209 patients affected with neurological disorders. The authors have analyzed the data for previously neurodevelopmental disorders-associated genes  and compared yield of the current cohort with previous studies of similar phenotypes and genetic studies. The study has also described different tools used to check minor allele frequency or the absence or presence of certain variants in control population, predictive programs, and decision support software. Furthermore, the study has highlighted the benefits and limitations of WES compared to whole genome sequencing. Overall, The article is well written, but I have a major concern regarding results of the study, as the authors have not added which of the variants were identified in which genes that could lead to the disease phenotypes. They have only shown the statistical analysis of the yield of the project in comparison to previous studies. The authors are suggested to add list of pathogenic and/or variants of uncertain significance in a supplementary table.

Minor comments:

There are several sentences that need correction e.g., 

Line 101-102: Written informed consent was obtained from all subjects prior to their participation. 

Some of the patients in the cohort are intellectually disabled, how could they provide consent for the study. I would suggest to take consent from the guardians/parents of such patients.

Line 102-103:  "DNA extraction was performed from peripheral blood following standard procedures." Add reference to show which of the methods was used for DNA extraction. 

Sentences at the following lines need correction:

Line 204-205

Line 233-235:

Line 281-284

Author Response

First of all, I would like to thank the reviewers for their comments. We have tried to answer all their questions, and we hope our answers are consistent with their suggestions. 

Response to Reviewer 1 Comments

1) I have a major concern regarding the results of the study, as the authors have not added which of the variants were identified in which genes could lead to the disease phenotypes. They have only shown the statistical analysis of the yield of the project in comparison to previous studies. The authors are suggested to add list of pathogenic and/or variants of uncertain significance in a supplementary table.

We have included a supplementary table with more information regarding clinical data and information of the detected variants.

2) There are several sentences that need correction e.g., 

Line 101-102: Written informed consent was obtained from all subjects prior to their participation. Some of the patients in the cohort are intellectually disabled, how could they provide consent for the study. I would suggest to take consent from the guardians/parents of such patients.

Line 101-102: we have included the sentence “or the guardians/parents of such patients” in the paragraph

Line 102-103:  "DNA extraction was performed from peripheral blood following standard procedures." Add a reference to show which of the methods was used for DNA extraction.

Line 102-103: we have added the methods used for DNA extraction

Sentences at the following lines need correction (Line 204-205, Line 233-235, Line 281-284)

We have corrected the paragraphs suggested for the reviewer 1

Reviewer 2 Report

In this manuscript, authors performed a retrospective evaluation of WES application in 209 cases referred to their hospitals, and discussed the impact of the implementation of WES as a routine tool for clinical diagnosis of neurological disorders. Their overall diagnostic rate of WES in patients with neurological diseases (NDs) is 32%, similar with the diagnostic yields reported by other groups in the context of clinical diagnostic testing. In addition, authors discussed some important facts regarding classification criteria for pathogenicity of rare variants, VUS and secondary findings. Overall, the manuscript is well written, but their presented data are somewhat limited. I have comments as follows:

1. Authough providing all the details for the 209 cases maybe difficult, authors are encouraged to share more information about their clinical data. At least a summary about autosomal dominant (de novo, inherited), X-linked, and autosomal recessive cases shall be presented.

2. Concerning the re-classification of the NIPBL variant (rs141976717), authors shall provide the clinical features of the proband. Were there any other possible candidate variants, de novo or bi-allelic, identified in the proband? If there are any other candidates, their classification (P, PP, VUS, PB or B) shall be presented together with results from predictive studies. If there are no other candidates, or other candidate variants are not known to be associated with the observed clinical features, their conclusion would be more convincing. In addition, did the authors consider the probability of mosaic of the variant in control databases that affect the penetrance? This may be worthy to discuss as well.

3. Table 1: Authors shall also include DGV as a commonly used database for genomic variants, and may consider adding “REVEL”, “CADD” as examples for “Predictive programs or in silico studies”.

4. Section 3.3. line 236-242: It’s worthy mentioning “pLI” and “missence Z score” here to explain the concept of gene constraint.

5. Section 3.7. WES limitations and new diagnostic strategies: Among these cases, are there any pathogenic variants (eg, non-coding variants, CNVs) that did not identified by WES but later solved by a new diagnostic strategy? If there are some, are the variants non-coding variants, CNVs, or exome variants? These details would provide more useful insights about how to choose diagnostic strategies.

6. Please also discuss that CNV detection using WES is still not decisive so far.

7. The overall diagnostic yield of WES varies from 30-50%, majority of the cases are remaining unsolved. Besides submitting the unsolved cases to GeneMatcher, retrospective re-analysis of undiagnosed cases can also increase the total diagnostic yield. What did the authors do for the unsolved cases? This maybe mentioned a bit more.

8. (Minor comment): Authors shall pay attention to the punctuation marks as some are missing in the manuscript.

Author Response

First of all, I would like to thank the reviewers for their comments. We have tried to answer all their questions, and we hope our answers are consistent with their suggestions. 

Response to Reviewer 2 Comments

1) Although providing all the details for the 209 cases may be difficult, authors are encouraged to share more information about their clinical data. At least a summary about autosomal dominant (de novo, inherited), X-linked, and autosomal recessive cases shall be presented.

We have included a supplementary table with more information regarding clinical data and information of the detected variants

2) Concerning the re-classification of the NIPBL variant (rs141976717), authors shall provide the clinical features of the proband. Were there any other possible candidate variants, de novo or bi-allelic, identified in the proband? If there are any other candidates, their classification (P, PP, VUS, PB or B) shall be presented together with results from predictive studies. If there are no other candidates, or other candidate variants are not known to be associated with the observed clinical features, their conclusion would be more convincing. In addition, did the authors consider the probability of mosaic of the variant in control databases that affect the penetrance? This may be worthy to discuss as well.

We have included more information regarding the NIPBL variant. Moreover in the paper Álvarez-Mora et al. 2022 (Reference 20), reviewer can find more information about this case.

3) Table 1: Authors shall also include DGV as a commonly used database for genomic variants, and may consider adding “REVEL”, “CADD” as examples for “Predictive programs or in silico studies”.

We have included databases and predictive programs suggested by reviewer in Table 1

4) Section 3.3. line 236-242: It’s worthy mentioning “pLI” and “missence Z score” here to explain the concept of gene constraint.

We have include a paragraph mentioning Z scores and pLI.

5) Section 3.7. WES limitations and new diagnostic strategies: Among these cases, are there any pathogenic variants (eg, non-coding variants, CNVs) that did not identified by WES but later solved by a new diagnostic strategy? If there are some, are the variants non-coding variants, CNVs, or exome variants? These details would provide more useful insights about how to choose diagnostic strategies.

Some of the cases have been included in research projects; nevertheless the paper is focused in the application of WES in the clinical practice. For example, before the development of new algorithms for CNV detection, we detect using WGS a deletion not detected by WES.

6) Please also discuss that CNV detection using WES is still not decisive so far.

A paragraph has been added discussing the detection of CNVs.

7) The overall diagnostic yield of WES varies from 30-50%, majority of the cases are remaining unsolved. Besides submitting the unsolved cases to GeneMatcher, retrospective re-analysis of undiagnosed cases can also increase the total diagnostic yield. What did the authors do for the unsolved cases? This maybe mentioned a bit more.

These patients have been included in research projects, nevertheless, in diagnostic labs, it is actually complicated to perform whole genome sequencing, methylation, or functional analysis.

8) (Minor comment): Authors shall pay attention to the punctuation marks as some are missing in the manuscript.

We have added some punctuation marks as suggested by the reviewer 2.

Reviewer 3 Report

In the present manuscript, The authors have discussed the use and implementation of Whole exome sequencing (WES) for clinical diagnostic purposes to identify Neurological Disorders. The authors have included the WES analysis of 209 cases they have performed. They also discussed the advantage and limitations of the WES. Overall the manuscript provides advancements in the current diagnostic methods for neurological disorders and should be accepted, however, some minor spell checks and reframing of the sentences are required. 

Author Response

First of all, I would like to thank the reviewers for their comments. We have tried to answer all their questions, and we hope our answers are consistent with their suggestions. 

 Response to Reviewer 3 Comments

 1) Some minor spell checks and reframing of the sentences are required.

We have corrected some minor spell checks and some sentences have been reframed 

Round 2

Reviewer 2 Report

The authors have addressed my comments and revised the manuscript accordingly. I don't have further concerns.